# Inhibitory Properties of Cinnamon Bark Oil against Postharvest Pathogen *Penicillium digitatum* In Vitro

**DOI:** 10.3390/jof10040249

**Published:** 2024-03-26

**Authors:** Ting Zhou, Jingjing Pan, Jingjing Wang, Qinru Yu, Pengcheng Zhang, Tongfei Lai

**Affiliations:** College of Life and Environmental Science, Hangzhou Normal University, Hangzhou 310036, China; zt20100061@hznu.edu.cn (T.Z.); 2022111010065@stu.hznu.edu.cn (J.P.); 2021111010050@stu.hznu.edu.cn (Q.Y.); zpc604@hznu.edu.cn (P.Z.)

**Keywords:** *Penicillium digitatum*, cinnamon bark oil, citrus fruit, transcriptome

## Abstract

*Penicillium digitatum* is a major postharvest pathogen that threatens the global citrus fruit industry and causes great economic losses annually. In the present study, inhibitory properties of cinnamon bark oil (CBO) against *P. digitatum* in vitro were investigated. Results indicated that 0.03% CBO could efficiently inhibit the spore germination, germ tube elongation, mycelial growth, colonial expansion and conidial accumulation of *P. digitatum*. The results of fluorescein diacetate (FDA) and MitoTraker Orange (MTO) staining also proved the suppression effects of CBO against *P. digitatum*. Meanwhile, CBO could inhibit green mold rots induced by *P. digitatum* in citrus fruit when the working concentration of CBO exceeded 0.06%. In addition, the expressions of 12 genes critical for the growth and virulence of *P. digitatum* were also significantly regulated under CBO stress. Through a transcriptomic analysis, a total of 1802 common differentially expressed genes (DEGs) were detected in *P. digitatum* after 4 h and 8 h of CBO treatment. Most of the DEG products were associated with carbohydrate, amino acid and lipid metabolism. They directly or indirectly led to the disturbance of the membrane and the generation of reactive oxygen species (ROS). Our results may deepen the understanding of antifungal properties of CBO against *P. digitatum* and provide the theoretical foundation to uncover the antifungal mechanism of CBO at the molecular level.

## 1. Introduction

Citrus fruits are among the most economically important fruit crops and are cultivated throughout the subtropical and tropical regions. In the postharvest handling, such as packaging, storage and transportation, many microorganisms can invade the fruits through mechanical wounds or biotic and abiotic factors that lead to fruit spoilage and substantial economic losses. Among them, *Penicillium digitatum* is a necrotrophic fungus that is the causal agent of green mold and accounts for most postharvest losses of citrus fruit [1]. The fungus has a short disease cycle and a strong sporulation capacity. Once the pathogen penetrates the host pericarp, it can quickly spread to the mesocarp and invade the adjacent cells by the germ tube. The infection leads to the breakdown of fruit cell walls and plasmolysis of pericarp and mesocarp cells, and results in a sunken soft watery spot. Then, the white mycelia and numerous greenish spores can be observed on the surface of the fruit lesion within a short time [2,3,4]. Meanwhile, *P. digitatum* can produce tryptoquialanines which are considered secondary metabolic toxins capable of eliciting sustained or intermittent tremors in vertebrate animals. These mycotoxins pose serious hazards to public health [5,6].

The application of synthetic fungicides (imazalil, pyrimethanil thiabendazole, prochloraz, fludioxonil, etc.) is the conventional management approach for controlling green mold disease [7]. However, the extensive use of fungicides will lead to environmental pollution, chemical residues in foods, and the development of resistant strains [8]. These deleterious impacts have been noticed and developing safe and efficient control approaches is warranted and urgent. So far, some physical treatments, such as heat shock treatment, ultraviolet light radiation, gamma ray irradiation and controlled atmosphere treatment have been practiced [9,10]. Nevertheless, the requirements for processing condition need to be satisfied, and the effects on fruit quality need to be evaluated. Biological control is one of alternative methods for minimizing postharvest fungal disease. Many well-known bacteria, yeasts, and a few fungi have shown protective and curative action in the control of green mold on citrus fruit [11,12,13]. The difficulties are how to balance the antagonistic efficiency against the target, survivability in adverse conditions, cultivation cost, compatibility with subsequent processing, and non-toxicity to hosts and consumers. In addition, some generally regarded as safe salts (GRAS) and food additives have demonstrated a high effectiveness against *P. digitatum* in vivo and in vitro [14,15,16]. Concerns regarding fruit quality parameters, dietary safety, finding a suitable combination with other physical treatments, and other regulatory issues still need to be addressed.

In recent years, many natural plant products including essential oils, natural compounds and plant extracts have exhibited a broad spectrum of activity against pathogens [17,18,19,20]. They are considered a promising natural alternative to synthetic fungicides due to their non-toxic, biodegradable, low-cost, and effective characteristics. Essential oils are commonly obtained from different plant materials and their active ingredients can vary based on their extraction sources such as roots, barks, leaves, flowers and fruits. Cinnamon (*Cinnamomum zeylanicum*) belongs to the Lauraceae family and is known popularly for its flavor and fragrance. Cinnamon bark has been used as a popular spice or a kind of medicinal substance by different cultures around the world for a long time [21]. Cinnamon bark essential oil (CBO) mainly consists of cinnamaldehyde, camphor, β-caryophyllene, eugenol, linalool, and other aliphatic and aromatic compounds [22]. Many reports indicated that CBO alone or combined with other strategies could postpone the senescence of postharvest fruits and vegetables, enhance the activities of defense-related enzymes to induce host defense responses, and suppress the occurrence of various postharvest diseases [23,24,25,26]. However, few data were available about the effects of CBO on the development of *P. digitatum*, and the antifungal mechanism of CBO was not entirely understood [27,28,29].

In the present study, the inhibitory effects of CBO against *P. digitatum* were investigated at physiological, biochemical and molecular levels. The transcriptomic changes in *P. digitatum* under CBO treatment were determined as well. The results will lead to a better understanding of the antimicrobial property and mechanism of CBO, and provide a theoretical basis for further developing the CBO-based postharvest control strategy.

## 2. Materials and Methods

### 2.1. Fungus, Fruit and Essential Oil

*P. digitatum* was isolated from a naturally infected citrus fruit (*Citrus reticulata* Blanco) with a typical green mold symptom, and the morphological, physiological and molecular characteristics were already determined in previous research. The pathogen was activated by inoculating in citrus fruit and maintained on potato dextrose agar (PDA) medium at 25 °C. The citrus fruit with commercial maturity were purchased from the local fruit market in Hangzhou, China. Cinnamon bark oil (CAS: 8015-91-6) was purchased from Merck KGaA (Darmstadt, Germany).

### 2.2. Antifungal Assays of CBO

The spores were collected from the sporulating cultures of *P. digitatum* by sterile water flooding and filtering with four layers of sterile gauze. Then, the proper amount of spore suspension was added to potato dextrose broth (PDB) medium with a final concentration of 1.0 × 10^6^ spores L^−1^. CBO was supplemented to the mixture with a final concentration of 0, 0.010, 0.015, 0.020, 0.025, 0.030 or 0.035% (*v*/*v*). Meanwhile, Tween-80 (Sangon, Shanghai, China) with a final concentration of 0.1% (*v*/*v*) was used to promote spore dispersion and facilitate the dissolving of CBO. After 4 to 12 h of culturing at 25 °C under 200 rpm shaking condition, the spore germination ratio and germ tube length of *P. digitatum* were determined by a Nikon DS-Fi1 microscope (Nikon, Tokyo, Japan). After 1 to 3 days of culturing with or without 0.03% CBO treatment, the mycelial dry weight of *P. digitatum* in a 100 mL system was measured after drying in a Modell 100–800 m oven (Memmert, Schwabach, Germany). To evaluate the effect of CBO on colony growth, a mycelial agar disk was placed in the center of the PDA plate containing 0 or 0.03% CBO. The colonial diameters were measured after 2, 4, 6 and 8 days of static culturing under 25 °C. To assess the effect of CBO on the conidial production of *P. digitatum*, 100 μL of fresh spore suspension (1.0 × 10^6^ spores L^−1^) was evenly spread on the PDA plate containing 0 or 0.03% CBO. After 3 to 12 days of static culturing under 25 °C, the spores were harvested with 10 mL of sterile water containing 0.1% Tween-80, and the spore number was counted using a Nikon DS-Fi1 microscope (Nikon, Tokyo, Japan) and a hemocytometer.

To evaluate the effect of CBO on green mold decay, the citrus fruits without mechanical injury were disinfected firstly by 2% sodium hypochlorite, and wounded at the fruit equator by a sterile nail. Then, 10 μL of the spore suspension (1.0 × 10^5^ spores L^−1^) with 0 or 0.06% CBO was injected into the wound. After 2 to 6 days of storing at 25 °C, the disease incidence and lesion size were measured, and the symptom of decay was photographed by a digital camera (Nikon, Tokyo, Japan). This experiment included three replicates and each treatment contained ten citrus fruits. The experiment was repeated once.

### 2.3. Fluorescence Staining

Fresh spores of *P. digitatum* were treated by 0 or 0.03% CBO in PDB medium for 4 h and 8 h at 25 °C under oscillating condition. According to the product instructions, the harvest spores were suspended in 20 mmoL L^−1^ phosphate buffer solution (pH 7.4), and stained by fluorescein diacetate (FDA) (Sangon, Shanghai, China) with a work concentration of 5μmoL L^−1^, MitoTraker Orange (MTO) (Invitrogen, Carlsbad, CA, USA) with a work concentration of 5 μmoL L^−1^, and propidium iodide (PI) (Sangon, Shanghai, China) with a work concentration of 20 mg L^−1^. The spores, which were treated with 0.03% CBO for 4 h and then incubated in boiling water for 10 min, were set to the positive control in the experiment of PI staining. Afterward, the spores were photographed by a fluorescence microscope (Eclipse Ni-U, Nikon, Tokyo, Japan).

### 2.4. Expression Analysis of Genes by qRT-PCR

Fresh spores of *P. digitatum* were treated by 0 or 0.03% CBO in PDB medium for 4 h, 8 h and 12 h at 25 °C under oscillating condition. After collection and removal residual media, total RNAs of spores and mycelia were extracted by the TRIzol Reagent (Invitrogen, Carlsbad, CA, USA), and cDNA templates were synthesized using a FastQuant RT Kit (Tiangen, Beijing, China) in strict accordance with the product description. The 2 × Ultro SYBR mixture (CW, Beijing, China) and a CFX96-real Time System (Bio-Rad, Hercules, CA, USA) were used for quantitative real-time PCR (qRT-PCR) detection. The information of designed primer pairs is shown in Appendix A. The relative expression levels of genes involved in the growth and virulence of *P. digitatum* were normalized by the *β-tubulin* gene and measured using the 2^(−ΔΔCt)^ analysis method.

### 2.5. Transcriptomic Analysis

Fresh spores of *P. digitatum* were treated by 0 or 0.03% CBO in PDB medium for 4 h and 8 h at 25 °C under oscillating condition. The harvested samples with a biological repeat (Pd4C-1, Pd4C-2, Pd4T-1, Pd4T-2, Pd8C-1, Pd8C-2, Pd8T-1 and Pd8T-2) were rinsed twice by sterile distilled water to remove residual media, quickly frozen in liquid nitrogen and sent to Beijing Genomics Institute Co., Ltd. (BGI, Beijing, China). Total RNA extraction and RNA sequencing (RNA-seq) were performed following a standard operating procedure of BGI (https://www.yuque.com/yangyulan-ayaeq/oupzan accessed on 11 February 2024). The process of total RNA preparation was described in Extraction of Microbial RNA BGI-NBS-TQ-RNA-002 A0. The cDNA library for each sample was constructed following the description of mRNA Library Preparation (DNBSEQ) BGI-NGS-JK-RNA-001 A0. Single-stranded circle DNA molecules were replicated via rolling cycle amplification, and a DNA nanoball (DNB) containing multiple copies of DNA was generated. On the BGISEQ-500 platform, sufficient quality DNBs were sequenced through combinatorial Probe-Anchor Synthesis (cPAS). The raw data were filtered with SOAPnuke (v1.5.2) and stored as FASTQ format. The clean data were mapped to *P. digitatum* Pd1 reference genome PdigPd1_v1 by HISAT (v2.1.0) and the assembled unique gene by Bowtie2 (v2.2.5). Then, the transcripts were annotated by the Non-Redundant Protein Sequence Database (NR), the Clusters of Orthologous Genes Database (COG), and the Kyoto Encyclopedia of Genes and Genomes (KEGG) database. The expression level of genes was calculated by RSEM (v1.2.8) and described as fragments per kilobase of the exon model per million mapped fragments (FPKM). Differentially expressed gene (DEG) analysis was conducted using DESeq software (fold change ≥ 2 and adjusted *p* value ≤ 0.001). A heatmap was drawn by pheatmap (v1.0.8) according to differential gene clusters. Furthermore, Gene Ontology (GO) (http://www.geneontology.org/ accessed on 11 February 2024) and KEGG (https://www.kegg.jp/ accessed on 11 February 2024) enrichment analysis of annotated DEGs were performed by Phyper (https://en.wikipedia.org/wiki/Hypergeometric_distribution accessed on 11 February 2024) with a Q value of ≤0.05 as the threshold.

### 2.6. qRT-PCR Validation

The expressions of six randomly selected DEGs were determined by qRT-PCR. Samples of *P. digitatum* were prepared in the same way as RNA sequencing, and the specific primer pairs were designed and listed in Appendix A. Total RNA extraction and first-strand cDNA synthesis, and qRT-PCR detection were performed as mentioned above.

### 2.7. Statistical Analysis

Except for specified notification, statistical analysis was performed by Microsoft Excel 2016. Data were pooled across at least three independent biological repeat experiments. Statistical significance was analyzed with Student’s *t*-test at each time point, and mean separations were analyzed using Duncan’s multiple range test. Differences at *p* < 0.05 were considered to be significant.

## 3. Results

### 3.1. Determining the Minimum Effective Concentration of CBO In Vitro

Spore germination and germ tube elongation of *P. digitatum* were determined under the different concentrations of CBO treatment. The results indicated that CBO showed obvious antifungal activity against *P. digitatum* in a dose-dependent manner. After 10 h culturing in the PDB medium, the spore germination rate of *P. digitatum* gradually decreased with CBO concentration going up. Compared with 82.34% in the control group, the germination rate reduced to 15.80% when the CBO concentration reached 0.03%. Meanwhile, the germ tube length in the 0.03% CBO-treated group was less than a quarter of that in the control group (Table 1). Therefore, the concentration of 0.03% was considered to be the minimum effective concentration of CBO against *P. digitatum* in vitro.

### 3.2. Inhibitory Effects of CBO on P. digitatum Growth

FDA can be hydrolyzed by intracellular esterases and produce fluorescein. The fluorescence intensity is positively correlated with cell vitality. After FDA staining, the control spores presented a strong green fluorescence, whereas the CBO-treated spores emitted a weak fluorescence (Figure 1A). Meanwhile, MTO can be used for staining mitochondria which generates most of the chemical energy needed to power the cell’s biochemical reactions. Under 0.03% CBO stress, the MTO-stained spores showed a weak or invisible red fluorescence that indicated a decrease in the number of mitochondria or a loss in mitochondrial membrane potential (Figure 1B). These results indirectly supported the assumption of the inhibition effects of CBO against *P. digitatum*.

In addition, compared with control, mycelial accumulation, colonial expansion and conidial production of *P. digitatum* were significantly suppressed by 0.03% CBO with increasing culture time (Figure 2A,B,D and Figure 3A). As a necrotrophic fungal pathogen, *P. digitatum* can effectively infect the citrus fruit through the wounds originating from mechanical damage. When the untreated conidia dispersed into the wound, *P. digitatum* initially germinated to produce germ tubes under suitable environmental condition, extended into mesocarp cells, and gradually invaded the adjacent cells. In the later infection process, white mycelia and newly generated grayish green conidia were observed on the surface and inside of the citrus fruit. The infected pericarp cells and mesocarp cells were plasmolyzed and their inclusions and organelles were coagulated, dark and digested. The citrus fruit initially showed a water logging symptom, then the rate of fruit deterioration progress accelerated as storage time went up 3 to 5 days, and the citrus fruit were finally fully rotted. In the preliminary experiment, the inhibitory effect on green mold disease in citrus fruits was not obvious when the concentration of CBO was less than 0.06%. Under 0.06% CBO treatment, all inoculated citrus fruits were infected by *P. digitatum,* and the phenotype also conformed to the typical disease symptom of green mold. However, the severity of green mold rot was obviously lower than that of control (Figure 2C, Figure 3B). After 6 days of inoculation, the lesion area of CBO-treated fruit was only half of the control, and the number of conidia covering the lesion surface were also significantly fewer than that of untreated fruit.

### 3.3. Effects of CBO on the Expression of Growth and Virulence-Related Genes

The expression changes of twelve genes involved in fungal development and virulence under CBO stress were detected. After 4 h of culturing, the expression levels of *PdMFS1*, *PdMF2*, *PdPMR1*, *PdMR5* and *PdSlt2* in the CBO-treated group were higher than those in the control group. The expression levels of *PdGCS1*, *PdPacC* and *PdCrz1* were down-regulated in the CBO-treated group. Otherwise, the expression of *PdMFS5*, *PdMpkB*, *PdMut3* and *PdMA1* were not influenced by CBO treatment. With the increase of CBO-treated time, the expression levels of all the genes, except for *Pdslt2*, were lower than those in the control group. Compared with the control, the expression of *PdSlt2* was significantly up-regulated (Figure 4).

### 3.4. Transcriptomic Changes of P. digitatum under CBO Treatment

The initiation of biochemical activities with an increase in metabolism and an induction of morphological changes of *P. digitatum* could be observed during 8 h of culturing. After 4 h of culturing, the control spores had been stimulated by the suitable environmental condition, and the vegetative growth was initiated with the swelling of cell volume. After 8 h of culturing, the spore germination and germ tube formation of untreated spores had been finished. Meanwhile, the development of CBO-treated spores was significantly inhibited. To explore the potential antifungal mechanism at the molecular level, a comparative transcriptomic analysis was performed on CBO-treated and untreated samples after 4 h and 8 h cultivation. Finally, more than 6.36 Gb data including at least 41.96 M clean reads were obtained for each sample, with the high-quality clean reads accounting for over 97.63%. The average genome mapping ratio and the mapping ratio with the reference gene were 94.27% and 70.65%, respectively. A total of 8256 genes including 8139 known genes and 117 novel genes were detected. Compared with control, 3625 (1919 up- and 1706 down-regulated) and 3055 (1596 up- and 1459 down-regulated) genes were differentially expressed in the CBO-treated group after 4 h and 8 h of culturing, respectively (Appendix A). The information of all DEGs is listed in Appendix A. In addition, 1802 DEGs were detected in both stages, and 1209 DEGs of them (731 up- and 478 down-regulated) showed the same expression trends at 4 h and 8 h of culturing (Appendix A and Figure 5). Compared with the results of qRT-PCR and transcriptome analysis, the expression trends of six randomly selected DEGs were basically consistent (Appendix A). The regression analysis results are shown in Figure 6. The Pearson correlation coefficient is 0.9705, indicating that the results of the two methods have a good correlation. Therefore, corresponding to the differences in fungal morphology, these DEGs are closely correlated with CBO treatment and play important roles in the growth and stress responses of *P. digitatum.*

GO analysis of 1802 DEGs indicated that the subcellular locations of most DEGs are cellular anatomical entities and protein-containing complexes. Most DEGs possess catalytic activity, binding activity, transporter activity or transcription regulator activity. They mainly take part in cellular processes, metabolic processes, and localization and biological regulation (Figure 7A). Results of KEGG pathway classification are shown in Figure 7B. Sorted by the number of DEGs involved, the top five pathways are carbohydrate metabolism, amino acid metabolism, lipid metabolism, metabolism of cofactors and vitamins, and translation. Further, GO enrichment analysis indicated that large amounts of DEGs are related to small molecule metabolic processes, organic acid metabolic processes, carbohydrate metabolic processes, carboxylic acid metabolic processes and oxoacid metabolic processes. DEGs associated with second-messenger-mediated signaling and fatty acid synthase activity showed the highest enrichment ratio (Figure 8A). Through KEGG pathway enrichment, three pathways of meiosis-yeast, cell cycle-yeast and peroxisome belong to the cellular processes catalog, whereas all other pathways belong to the metabolism catalog. Among them, seven pathways containing a relatively large number of DEGs are involved in carbohydrate metabolism, such as starch and sucrose metabolism, amino sugar and nucleotide sugar metabolism, glyoxylate and dicarboxylate metabolism, pyruvate metabolism (Figure 8B). And five pathways are associated with the metabolism of secondary metabolites. Therefore, CBO treatment had a clear effect on the complex and highly regulated enzymatic machinery which is involved in the metabolism of macromolecules (especially structural and storage carbohydrates) and secondary metabolites of *P. digitatum*.

## 4. Discussion

Green mold is an economically significant postharvest fungus that mainly hazards citrus fruit and derivatives worldwide. Conventional synthetic fungicides have been questioned due to the potential deleterious impacts on human health and the environment. Therefore, searching for safe, effective and eco-friendly approaches is necessary and urgent. Using essential oils to control postharvest diseases is gaining importance because of their highly antimicrobial, antioxidant, non-toxic and low residual attributes. For example, essential oils of oregano, fennel, peppermint, laurel, rosemary, lemon grass, eucalyptus, clove and neem were strongly inhibitory to growth of *P. digitatum* in vitro and in vivo [1,30,31]. CBO is categorized as generally recognized as a safe (GRAS) compound and has excellent antimicrobial properties. In this study, the inhibitory effects of CBO on the growth of *P. digitatum* were first determined. The experimental results showed that 0.03% CBO could efficiently suppress the spore germination, germ tube elongation, mycelial accumulation, colonial expansion and conidial production. These results are consistent with the fluorescence staining results of FDA and MTO. In addition, with an increase in CBO concentration, the green mold rots in citrus fruits induced by *P. digitatum* were substantially controlled. These results can be a reference for easy, cost-effective and environment-friendly management and control of green mold rot in postharvest citrus fruit with CBO. To further explore the application potential of CBO, one of following goals is to optimize the extraction process parameters of CBO with the aim of maximizing the yield while retaining quality, the other is to improve antifungal, physical and sustained-release properties of CBO derivatives by performing CBO vapor fumigation, preparing CBO-microcapsules or microemulsions, or developing an edible coating enriched with CBO.

It is widely agreed that the antifungal activity of CBO is caused by various effects on different cell components rather than a unique mechanism of action, such as inhibiting ATPases and cell division, altering the lipid profile, damaging cell membrane, and anti-quorum sensing effects [32]; Zhang et al. (2021) found that cinnamon essential oil-mesoporous silica nanoparticles induce a large number of reactive oxygen species (ROS), thereby inhibiting spore germination and growth of *Mucor* sp. [33]. Our previous study demonstrated that CBO could disturb carbohydrate metabolic process of *P. expansum* [34]. Lee et al. (2020) revealed that the antifungal activity of CBO against *Raffaelea quercus-mongolicae* and *R. solani* was due to ROS generation and cell membrane disruption [35]. He et al. (2018) showed that CBO could pass through the cell wall and the plasma membrane, and interact with the membranous structures of cytoplasmic organelles of *Colletotrichum acutatum* [36]. Darvishi et al. (2013) found that eugenol in CBO could interfere with Tat1p and Gap1p permeases which were related to dual transport of aromatic and branched-chain amino acids through the cytoplasmic membrane of yeast [37]. Nevertheless, the membranes of *P. digitutam* spores were still integrated after 4 h of culturing under 0.03% CBO treatment, which was not consistent with previous reports (Appendix A). A similar situation occurs with *Escherichia coli* under trans-cinnamaldehyde treatment with a sub-lethal concentration [38]. The most likely explanations are related to differences in the fungal species or the use of a lower concentration of CBO.

Furthermore, the effects of CBO on expression levels of several key genes related to the growth and virulence of *P. digitatum* were also examined. With increasing CBO-treatment time, all genes demonstrated a lower level of expression than that in control, while *PdSlt2* displayed a higher level of transcription. The major facilitator superfamily (MFS) and the ATP-binding cassette superfamily (ABC) comprise many important secondary transporters and play different roles during pathogen-fruit interaction. The MFS and ABC transporters in phytopathogenic fungi, excepting MFS5, contribute to increasing fungal aggressiveness by transporting a broad spectrum of substrates and granting multidrug resistance to fungi [39,40,41]. Therefore, the transcription of *PdMFS1*, *PdMFS2*, *PdPMR1* and *PdPMR5* in *P. digitatum* were induced by CBO treatment at first, whereas the expression of *PdMFS5* was unaffected. The *Slt2* mitogen-activated protein is a positive regulator of two sterol demethylases and a negative regulator of several MFS and ABC. *PdSlt2* MAPKs were generally implicated in responding to environmental stress, maintaining cell wall integrity and regulating secondary metabolite production [42]. In this context, the elevated level of *PdSlt2* was observed in *P. digitatum* under CBO stress. Glucosylceramides synthase (Gcs1) can transfer glucose group to ceramides and regulate the physical properties of the membrane [43]. The *P. digitatum* MAPK kinase (PdMpkB) is negatively correlated with osmotic stress adaptation and regulates the genes involved in cell wall-degrading enzyme activities, carbohydrate and amino acid metabolisms [44]. PdMut3, a Zn_2_Cys_6_ transcription factor, is not associated with fungicide sensitivity, but has an indirect effect in *P. digitatum* virulence through metabolism and peroxisome development [45]. The plasma membrane H^+^-ATPase (PMA1) could pump protons from the cytosol that coupled with hydrolyzing ATP, which played an important role in maintaining pathogenesis in *P. digitatum* [46]. The pH signaling transcription factor PacC can regulate the expressions of genes related to cell wall degradation enzymes, and is necessary for full virulence in *P. digitatum* [47]. The calcineurin-responsive transcription factor Crz1 could regulate membrane lipid homeostasis and is required for sporulation, full virulence and 14α-demethylation inhibitor resistance in *P. digitatum* [48]. These genes play regulatory functions in growth, physiological metabolism and fungal virulence in *P. digitatum*, and are appreciably affected by CBO treatment.

In addition, the high-quality genome sequencing of *P. digitatum* has been completed, which provides an optimum resource for understanding the molecular basis of pathogenicity and fungicide resistance formation of this pathogen [49,50,51]. Based on the whole genome information, transcriptomics can comprehensively and intuitively display the gene expression of pathogenic fungi in the process of infection. Thus, many interspecies or intraspecies comparative transcriptomic analyses have been performed to investigate the driving forces of fungal host switches and effectors functioning in plant–pathogen interactions [52,53]. The inhibitory mechanisms of many novel antifungal materials were also revealed by omics approaches. For examples, with the help of transcriptional profiling analysis, OuYang et al. (2016) found that citral exposure affected the expression levels of five ergosterol biosynthetic genes (ERG7, ERG11, ERG6, ERG3 and ERG5), led to the reduction in ergosterol content, and induced accumulation of massive lanosterol in *P. digitatum* [54]. Feng et al. (2020) indicated that a total of 938 DEGs were detected in peptide thanatin-treated *P. digitatum*, and the underlying mechanism might be the genetic information processing and stress response [55]. Lin et al. (2020) found that the X33 oligopeptide produced by *Streptomyces lavendulae* could affect energy metabolism, oxidative stress, and transmembrane transport, and then inhibit the hyphae polarization of *P. digitatum* [56]. The transcriptional and metabolome profiling also revealed 3648 DEGs and 190 prominently changed metabolites, which suggested that X33 oligopeptide mainly inhibited *P. digitatum* growth by affecting cell integrity, genetic information delivery, oxidative stress tolerance, and energy metabolism [57]. Yang et al. (2021) reported that, according to RNA-seq analysis, nanoemulsion containing cinnamaldehyde, eugenol or carvacrol mostly affected cellular respiration, proton transmembrane transport and guanyl nucleotide-binding of *P. digitatum*. The metabolic pathway, biosynthesis of secondary metabolites, and glyoxylate and dicarboxylate metabolism were disturbed as well under nano-emulsion stress [58]. Transcriptomic analysis also showed that the interference with ribosome, genetic information processing, cell membrane metabolism and energy metabolism might be closely involved in the antifungal mechanism of sodium dehydroacetate against *P. digitatum* [59]. Recently, through transcriptomic study, researchers suggested antifungal protein AfpB contributed to the overall homeostasis of the cell, repressed toxin-encoding genes, and linked with apoptotic process of *P. digitatum* [60].

In the present study, the transcriptomic analysis was also conducted to profile DEGs in CBO-treated *P. digitatum*, which helped us to promote a better understanding of the specific antifungal mechanisms of CBO. Unlike transcriptomic patterns of other antifungals exposure, a total of 3625 and 3055 DEGs in *P. digitatum* were identified after 4 h and 8 h of CBO treatment. Among them, 1802 genes were differentially expressed in both groups, which were more likely to highlight the effects of CBO at the molecular level. As direct or indirect targets of CBO, these DEGs are mostly located in cellular anatomical entities and protein-containing complexes, possessed binding and catalytic activities, and mainly participated in the cellular processes, localization, metabolic processes, biological regulation and response to stimulus. This is attributable to lipid solubility and permeability across living cell membranes, and is consistent with the broad-spectrum antimicrobial effect and multiple antifungal mechanisms of CBO. Through KEGG pathway classification, the number of DEGs related to carbon source metabolism (including carbohydrate, amino sucrose, nucleotide sugar, glyoxylate and dicarboxylate, pyruvate and pentose phosphate) was the highest, and the enrichment ratio of involved pathway was higher as well. Carbohydrates and their derivatives have a significant effect on energy reservoir and conversion, protein regulation and modification, nucleic acid skeleton, and principal structural components [61]. Lai et al. (2021) also reported that cinnamon oil could disturb the carbohydrate metabolic process in *P. expansum* at the protein level [34]. In addition, 776 DEGs are related to amino acid metabolism. Amino acids are generally used in synthesis of proteins and a variety of physiologically active nitrogenous compounds (enzymes, hormones, and other functional substances). They are the main source of nitrogen and an alternative energy source instead of carbohydrates and fats [62,63]. Predictably, CBO treatment also leads to the imbalance of intracellular nitrogen, disturbance of energy metabolism, and disorder of protein metabolism by interfering with amino acid metabolism. The amount of DEGs involved in translation, energy metabolism, folding, sorting and degradation of proteins have offered some support for this view. Meanwhile, 365 DEPs belong to lipid metabolism, that is the process by which fatty acids are digested, broken down or stored for energy. After enrichment analysis, DEGs associated with fatty acid synthase activity and glycerophospholipid metabolism were noticed. Glycerophospholipids are fatty acid diglycerides with a phosphatidyl ester attached to the terminal carbon, they are the major structural lipid components of eukaryote cellular and vesicle membranes [64,65]. Many reports have shown that CBO could affect the integrity of cell membranes by damaging cell-membrane components and resulting in osmotic disturbance. Abnormal expressions of lipid metabolism-related DEGs in CBO exposed *P. digitatum* might be considered a stress response or a compensatory mechanism to overcome the cytotoxicity of CBO. The peroxisome is a cytoplasmic organelle which acts in oxidative reactions and especially in the production and decomposition of hydrogen peroxide [66]. A total of 11 DEGs and 43 DEGs related to the peroxisome were highlighted in GO and KEGG enrichment analysis, respectively. This matches well the reactive oxygen species mediated-antifungal activity of CBO. Furthermore, six DEGs take part in the vitamin B6 metabolism and 26 DEGs contribute to vitamin B6 binding. Vitamin B6 is a generic term referring to the ensemble of several interconvertible pyridine compounds. Its primary role is acting as an enzymatic cofactor in an enormous variety of biochemical transformations (especially in protein or amino acid metabolism). It is also a potent antioxidant that effectively quenches ROS and is thus of high importance for cellular well-being [67,68]. The transcription level changes of DEGs in vitamin B6 metabolism upon CBO treatment echo other results in the present study and are in line with previous reports. Through transcriptional profiling and bioinformatics analysis, the possible key genes, metabolic pathways responding to CBO treatment were identified. Further function analysis and correlation of these candidates by the actual experiments will help to discover the sophisticated and complex antifungal mechanism of CBO at the molecular levels.

## 5. Conclusions

In the present study, CBO inhibited the growth of *P. digitatum* in a dose-dependent manner in vitro. It can delay the progression of green mold rot on citrus fruits. The expressions of 12 genes critical for the growth and virulence of *P. digitatum* were also significantly regulated with increasing CBO-treatment time. Furthermore, transcriptome analysis indicated that CBO treatment mainly induced the disturbance of carbohydrate, amino acid and lipid metabolism in *P. digitatum*. Our results may deepen the understanding of the antifungal molecular mechanism of CBO against *P. digitatum*.

## Figures and Tables

**Figure 1 jof-10-00249-f001:**
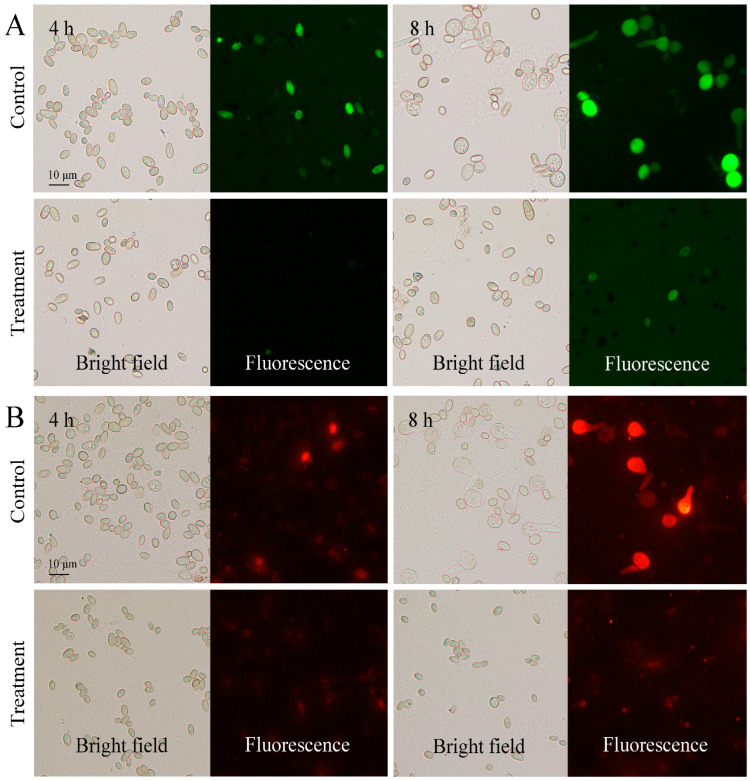
Fluorescein diacetate (**A**) and MitoTrack Orange (**B**) staining of *P. digitatum* spores with or without 0.03% CBO treatment after 4 h and 6 h of culturing.

**Figure 2 jof-10-00249-f002:**
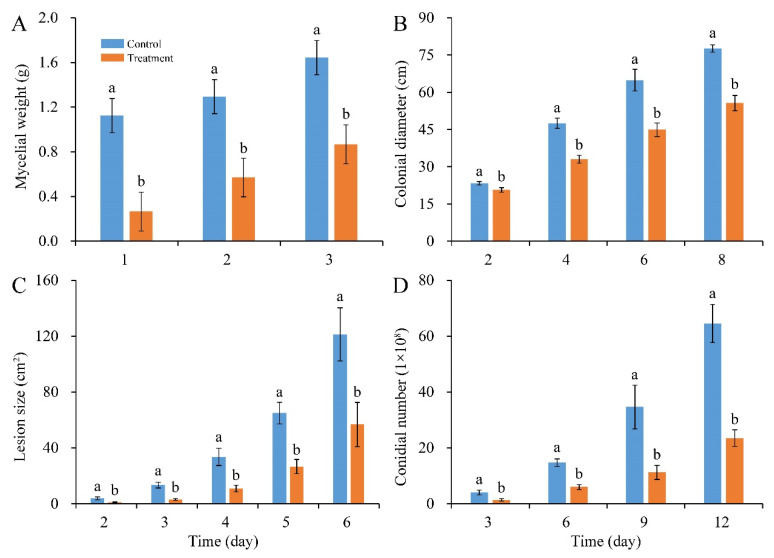
Inhibitory effects of CBO on *P. digitatum* growth and green mold rot on citrus fruit. (**A**) mycelial weight; (**B**) colonial growth; (**C**) lesion expansion on citrus fruit; (**D**) conidial production. Bar represents the standard deviation of the means of three independent experiments. Lowercase letters indicate significant differences at *p* < 0.05 at each time point.

**Figure 3 jof-10-00249-f003:**
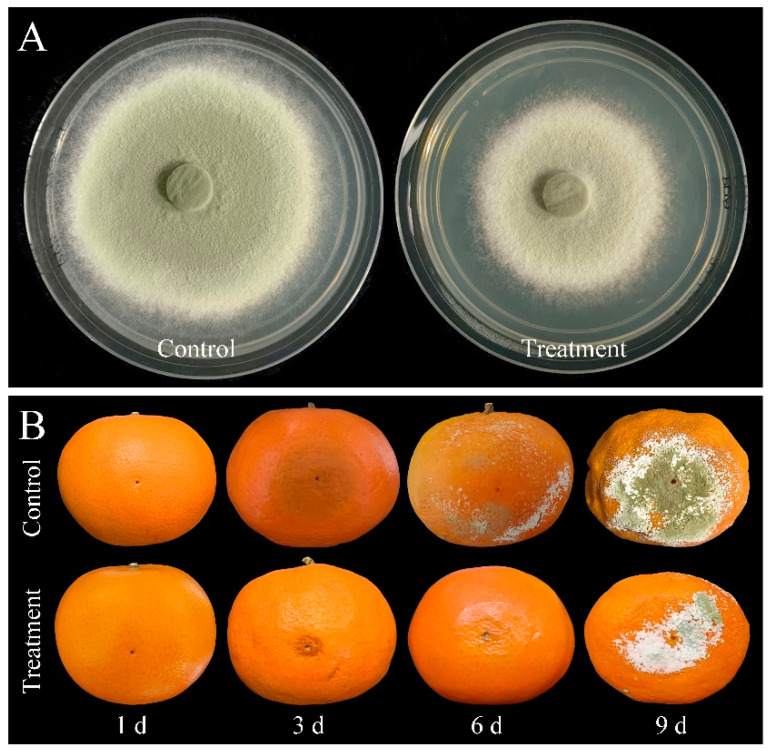
(**A**) The colonial morphology of *P. digitatum* under 0.03% CBO treatment in PDA medium after 8 days of culturing. (**B**) Symptoms of citrus fruits inoculated by *P. digitatum* with or without 0.06% CBO treatment.

**Figure 4 jof-10-00249-f004:**
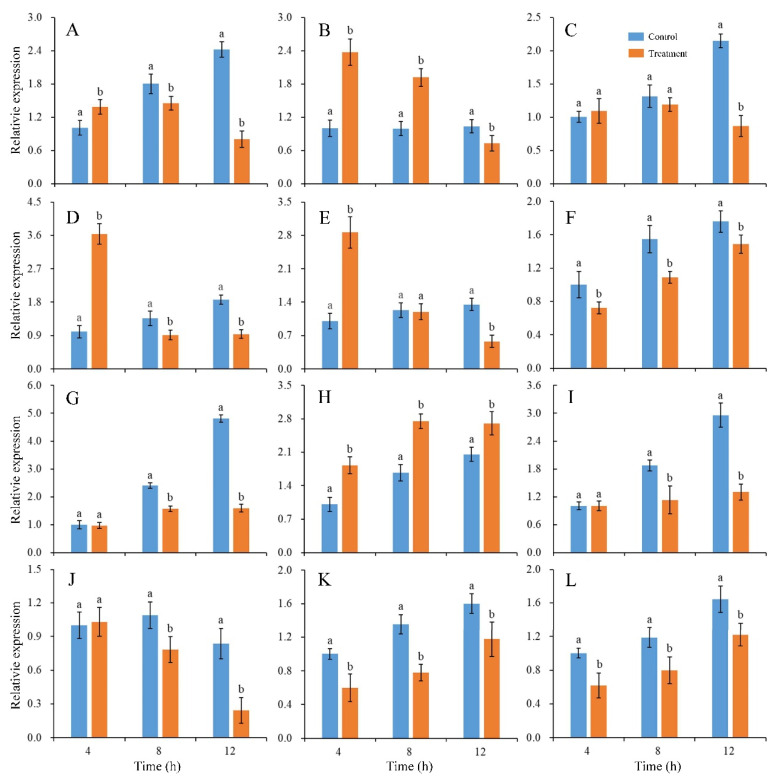
Effects of CBO on expressions of genes involved in growth and virulence of *P. digitatum*. Expression levels of genes are determined by qRT-PCR at the indicated time. The *β-tubulin* housekeeping gene is used as the internal control. Bar represents the standard deviation of the means of three independent experiments. Lowercase letters indicate significant differences at *p* < 0.05 at each time point. (**A**–**L**) represent the expression of *PdMFS1*, *PdMFS2*, *PdMFS5*, *PdMR1*, *PdMR5*, *PdGCS1*, *PdMpkB*, *PdSlt2*, *PdMut3*, *PdMA1*, *PdPacC*, and *PdCrz1*, respectively.

**Figure 5 jof-10-00249-f005:**
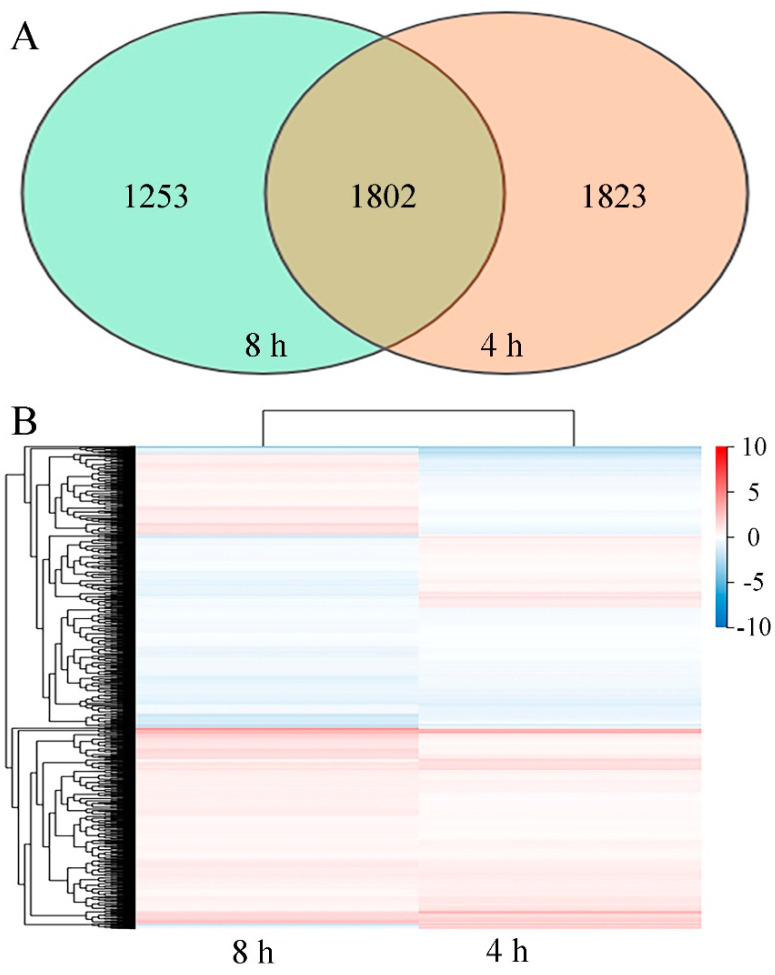
(**A**) Venn diagram showing the overlap of DEGs from 4 h and 8 h datasets. (**B**) Heatmap of DEGs in *P. digitatum* under 0.03% CBO treatment after 4 h and 8 h of culturing. Gradient color barcode indicates log_2_(FC) value (FC, Fold change of expression in control case to expression in CBO-treated case). Each column indicates a certain time point and each row indicates a DEG. DEGs with similar fold change values are clustered both at row and column levels.

**Figure 6 jof-10-00249-f006:**
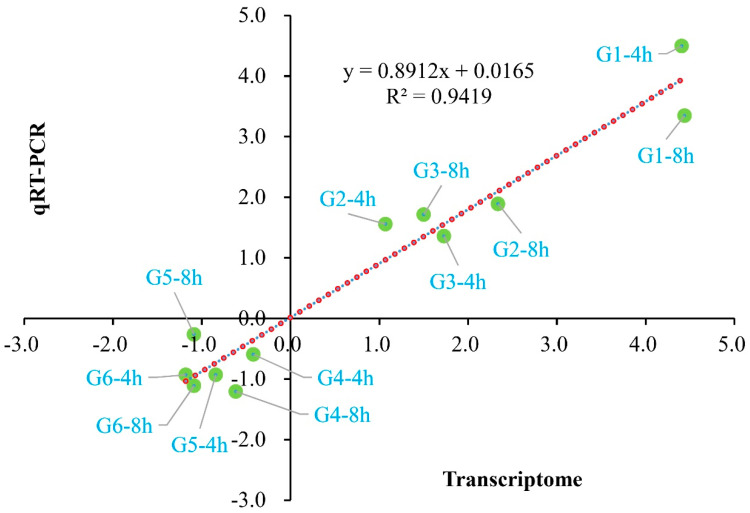
Regression analysis of randomly selective DEGs relative expression exposed by qRT-PCR and transcriptome sequencing. The X axis indicates log_2_Ratio(FC) acquired from the transcriptome sequencing method, and the Y axis indicates log_2_Ratio(FC) acquired from the qRT-PCR method. The green dots represent the different DEGs and the red dotted line indicates the optimum imitative straight line. The R indicates the correlation coefficient. The information about G1 to G6 is listed in Appendix A.

**Figure 7 jof-10-00249-f007:**
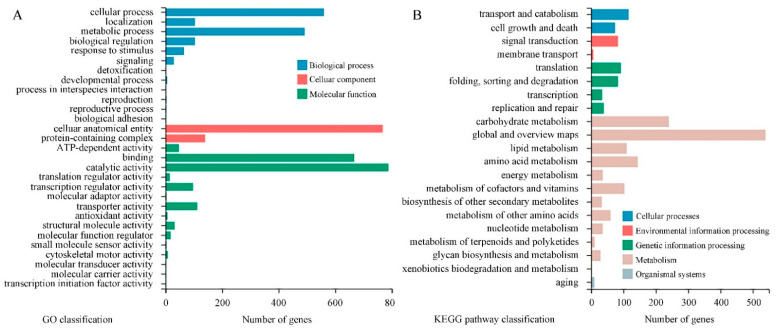
The GO (**A**) and KEGG (**B**) classification of 1082 DEGs. The X axis indicates the DEGs number and the Y axis indicates different GO or KEGG pathway terms. All secondary terms are grouped into primary terms with different colors.

**Figure 8 jof-10-00249-f008:**
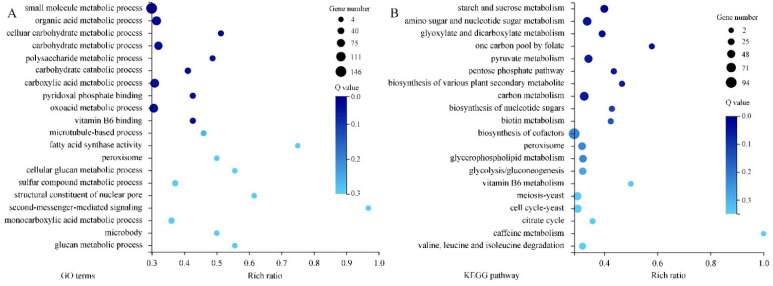
The GO (**A**) and KEGG (**B**) enrichment of 1082 DEGs. The rich ratio is the ratio of target gene numbers annotated in this pathway term to all gene numbers annotated in this pathway term. The greater rich ratio value indicates the greater the degree of enrichment. The dot represents the number of DEGs and the larger size indicates the greater number. The Q-value is the corrected *p*-value, and the lower Q-value indicates the greater enrichment level.

**Table 1 jof-10-00249-t001:** Effects of CBO on spore germination and germ tube elongation.

Treatment (%)	4 h	6 h	8 h	10 h	12 h
	Spore Germination (%)				
0	4.33 ± 1.22 a	36.62 ± 2.68 a	61.51 ± 3.96 a	82.34 ± 6.04 a	89.02 ± 2.52 a
0.010	2.43 ± 0.66 b	28.50 ± 4.17 b	37.51 ± 6.87 b	44.12 ± 3.75 b	58.29 ± 13.38 b
0.015	1.80 ± 1.01 bc	26.10 ± 3.94 bc	33.20 ± 3.73 b	42.61 ± 6.03 bc	55.08 ± 5.93 b
0.020	1.05 ± 0.41 c	18.35 ± 4.49 c	27.57 ± 6.72 bc	38.28 ± 6.78 bc	47.75 ± 5.32 b
0.025	0.91 ± 0.42 c	6.00 ± 1.97 d	18.42 ± 3.09 c	33.20 ± 4.27 c	43.87 ± 4.56 b
0.030	0.74 ± 0.23 c	1.73 ± 0.60 e	4.26 ± 1.49 d	15.80 ± 2.74 d	18.73 ± 4.62 c
0.035	0.67 ± 0.44 c	1.30 ± 0.43 e	1.75 ± 0.78 d	5.42 ± 1.51 e	12.37 ± 6.53 c
	**Germ tube length (μm)**				
0	4.38 ± 2.82 a	27.73 ± 6.48 a	47.80 ± 11.72 a	94.70 ± 16.98 a	253.05 ± 83.76 a
0.010	3.53 ± 1.36 a	19.03 ± 5.45 ab	33.36 ± 9.25 ab	77.94 ± 10.95 a	169.16 ± 46.30 ab
0.015	3.60 ± 1.20 a	19.50 ± 4.17 ab	28.75 ± 7.81 ab	67.64 ± 10.67 ab	120.12 ± 34.60 bc
0.020	3.94 ± 0.90 a	12.88 ± 4.66 b	26.87 ± 6.33 b	48.56 ± 11.14 b	92.56 ± 23.60 c
0.025	2.87 ± 1.17 a	5.36 ± 2.51 c	19.07 ± 6.49 b	39.44 ± 6.46 b	68.81 ± 19.40 c
0.030	3.54 ± 0.80 a	3.59 ± 1.44 c	6.50 ± 2.89 c	22.03 ± 5.52 c	36.76 ± 5.70 d
0.035	2.64 ± 0.81 a	2.28 ± 0.93 c	4.14 ± 2.02 c	15.00 ± 4.33 c	25.50 ± 5.93 d

Each value is the mean of three independent experiments and the standard deviation. Lower case letters indicate significant differences between treatments at *p* < 0.05 at the indicated time.

## Data Availability

The data used to support the findings of this study can be made available by the corresponding author upon request.

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
