# Peer review of "Inhibitory Properties of Cinnamon Bark Oil against Postharvest Pathogen *Penicillium digitatum* In Vitro"

_jof, 2024, doi:10.3390/jof10040249_

Round 1
Reviewer 1 Report
In general, the work provides information on the use of cinnamon essential oil against penicillium both, at the in vitro and molecular level and an explanation of the possible modes of action of the essential oil. The information is interesting but I believe that, from a post-harvest perspective, it would be important to include what happened to the fruit after puncturing.
Being a post-harvest study, I would have liked to know what happened inside the fruit.
Furthermore, I have doubts regarding the industrial usefulness of the puncture process carried out.
Indicate what happened to the fruit, inside, once the puncture was performed.
Clarify if the fruit should be injected in several places to avoid the spread of the fungus.
Author Response
Dear reviewer,
First of all, we would like to thank you very much for giving us such good comments about this paper. We have earnestly revised this paper, according to the comments and suggestions made by you, and decide to submit the revised manuscript, in which the specific revisions have been highlighted in bright yellow, to Journal of Fungi. A detailed explanation of how we have dealt with the points raised by the reviewers can be found in attachment.
Finally, we would like to thank you again and we are looking forward to hearing your reply.
Best wishes!
Yours sincerely,
Tongfei Lai
On behalf of all the authors
College of Life and Environmental Science, Hangzhou Normal University, Hangzhou 310036, China

Reviewer 2 Report
The manuscript "Inhibitory properties of cinnamon bark oil against postharvest pathogen Penicillium digitatum in vitro" by T. Zhou et al. is devoted to the development of effective antifungal treatment of citrus fruits against green mold desease. The antimicrobial action of cinnamon oil on pathogenic Penicillum digitatum was characterized using micro- and macroscopic analysis, transcriptomics.
I gues the presented manuscript could be accepted for publication after minor revision. Some needed corrections I've highlighted here. As well as I should to note that transcriptomics results disscussion is not so informative without the comparative data for other antifungals (fungicidal or fungistatic). Does the cinnamon oil treatment resembles the same pattern or not? The main conclusions from authors and literature data should be summarized in the text, it will improve the quality of the presentation.
ref. 52: "Molecules" should be itallic
Author Response

(The authors gave the same response as above.)
